# Application of Proteomics in Pancreatic Ductal Adenocarcinoma Biomarker Investigations: A Review

**DOI:** 10.3390/ijms23042093

**Published:** 2022-02-14

**Authors:** Christina Jane Vellan, Jaime Jacqueline Jayapalan, Boon-Koon Yoong, Azlina Abdul-Aziz, Sarni Mat-Junit, Perumal Subramanian

**Affiliations:** 1Department of Molecular Medicine, Faculty of Medicine, Universiti Malaya, Kuala Lumpur 50603, Malaysia; christinajane98@gmail.com (C.J.V.); azlina_aziz@um.edu.my (A.A.-A.); sarni@um.edu.my (S.M.-J.); 2University of Malaya Centre for Proteomics Research (UMCPR), Universiti Malaya, Kuala Lumpur 50603, Malaysia; 3Department of Surgery, Faculty of Medicine, Universiti Malaya, Kuala Lumpur 50603, Malaysia; bkyoong@gmail.com; 4Department of Biochemistry and Biotechnology, Annamalai University, Chidambaram 608002, Tamil Nadu, India; psub@rediffmail.com

**Keywords:** pancreatic ductal adenocarcinoma, biomarkers, CA 19-9, proteomics, diagnosis, prognosis, monitoring of treatment response, tumour recurrence

## Abstract

Pancreatic ductal adenocarcinoma (PDAC), a highly aggressive malignancy with a poor prognosis is usually detected at the advanced stage of the disease. The only US Food and Drug Administration-approved biomarker that is available for PDAC, CA 19-9, is most useful in monitoring treatment response among PDAC patients rather than for early detection. Moreover, when CA 19-9 is solely used for diagnostic purposes, it has only a recorded sensitivity of 79% and specificity of 82% in symptomatic individuals. Therefore, there is an urgent need to identify reliable biomarkers for diagnosis (specifically for the early diagnosis), ascertain prognosis as well as to monitor treatment response and tumour recurrence of PDAC. In recent years, proteomic technologies are growing exponentially at an accelerated rate for a wide range of applications in cancer research. In this review, we discussed the current status of biomarker research for PDAC using various proteomic technologies. This review will explore the potential perspective for understanding and identifying the unique alterations in protein expressions that could prove beneficial in discovering new robust biomarkers to detect PDAC at an early stage, ascertain prognosis of patients with the disease in addition to monitoring treatment response and tumour recurrence of patients.

## 1. Introduction

Pancreatic cancer (PanC) is an aggressive malignancy of the digestive and endocrine system that develops in the head of the pancreas most commonly, as well as in the tail and body of the organ [1]. The majority of PanC arises from exocrine glands of the pancreas, in which pancreatic ductal adenocarcinoma (PDAC) is the most common type, while the less common pancreatic tumours are of the endocrine type (e.g., pancreatic neuroendocrine tumour (PNET) [2] (Figure 1)).

The incidence of PanC has increased worldwide in recent decades and is expected to continually rise [3,4,5,6,7]. Moreover, PanC is the seventh leading cause of mortality by cancer worldwide [8]. Globally, the incidence and mortality rate of PanC is observed at 4.9% and 4.5%, respectively [9]. In the Malaysian context, the incidence and mortality rate of PanC is rather low (2.2% and 3.6%, respectively) as compared to other Asian countries such as Singapore (3.4% and 6.2%, respectively) and Japan (4.3% and 9.6%, respectively) as well as worldwide [9]. Regardless, its infamous reference as a silent killer steadfastly remains with its aggressive clinical symptoms only being presented at an advanced stage, thus posing a great challenge for intervention at the early stage of disease [10]. Further, unlike cancers of the colorectum [11], breast [12] and lung [13] which was reported to have a reduced mortality rate due to the advancement in their treatment modalities, limited options available for the treatment of patients with advanced PanC confer only a minimal effect to patients’ overall survival [14]. Given this, a myriad of research is being conducted worldwide to develop effective biomarkers for PanC that aids in the early detection of the disease as well as to evaluate prognosis and monitor treatment response of PanC patients, which in turn could make room for unrestricted treatment and management options, in addition to improvement of its current dismal 5-year survival rate [15,16,17,18].

The term ‘biomarker’ refers to the characteristics or signs that are measurable and as such could indicate the normal or pathogenic biological processes or responses to treatment [19]. Along this line, a diagnostic biomarker detects and confirms the presence of disease while a prognostic/predictive biomarker identifies the outcomes of disease and/or tumour recurrence. On the other hand, treatment response biomarkers assess whether a particular treatment is beneficial to an individual.

There are numerous strategies exploited for the exploration and/or identification of biomarkers for PanC [20]. There are those that are based on various DNA, mRNA, microRNA (miR), small nuclear RNA, long noncoding RNA, proteins, circulatory tumour cells, metabolites, and carbohydrates (glycans) (Table 1) using different types of biological sample matrices. The development of cancer is a multistep process including various alterations of protein structures, functions, interactions, and expressions [21]. In light of this, proteomics has emerged as one of the popular methodologies for PanC research [22]. Hence, in the present review, we aim to discuss the recently published literature that focuses on protein biomarkers discovered via various proteomic approaches intended for potential use as diagnostic, prognostic, treatment response and/or tumour recurrence markers for PDAC.

### 1.1. PDAC: Risk Factors, Diagnosis, Staging and Treatment

PDAC typically appears as poorly defined masses with extensive fibrosis surrounding the tumour tissues known as desmoplasia. In PDAC, desmoplasia promotes the growth of the tumour and metastatic spread as well as inhibition of drug penetration and uptake [55]. The presence of extensive desmoplastic stroma consisting of pancreatic stellate cells, proliferating fibroblasts, inflammatory cells and macrophages, marrow-derived stem cells and various other growth factors (e.g., epidermal growth factor, fibroblast growth factor, transforming growth factor-β), all of which are suitable for nourishing and facilitating invasive behaviour of the cancer cells, making it a distinguishing feature for PDAC [56].

The development of PDAC is not de novo. It could originate from three types of precursor lesions which include mucinous cystic neoplasm, intraductal papillary mucinous neoplasm and pancreatic intraepithelial neoplasia (PanIN), with higher predominance [57]. Individuals with chronic pancreatitis, diabetes, a family history of PDAC, genetic disorders (e.g., Lynch syndrome and Peutz–Jeghers syndrome) as well as poor lifestyle habits are generally at higher risk of developing this disease [8].

At present, the gold standard for the diagnosis of PDAC is an endoscopic ultrasound (EUS)-guided fine-needle aspiration biopsy (FNAB) [58] followed by histopathological examination (HPE) of the tumour tissue for microscopic characterisation and assessment [59]. Further, EUS imaging supports the diagnosis of PDAC based on the visualization of tumour size (>2 cm), vascularity of the tumour mass (irregular arterial and/or absence of venous vasculature), irregular dilation of the pancreatic ducts, absence of cysts within the tumour mass and presence of lymphadenopathy [58]. Other imaging modalities used in the diagnosis of PDAC include non-invasive imaging such as magnetic resonance imaging (MRI) and multidetector computed tomography (CT) [60], and minimally invasive imaging such as endoscopic retrograde cholangiopancreatography [61].

Despite its wide use in clinics, these imaging techniques are not without limitations [62]. For example, EUS is not effective in differentiating malignant lesions from those of inflammatory masses, thus complicating treatment decisions particularly among the latter conditions [63]. Further, FNAB is generally not recommended for body and tail tumours of the pancreas (PDAC) during EUS procedures due to needle tract seeding, thus posing a theoretical risk of spread through the biopsy needle [64]. CT and MRI imaging, on the other hand, can easily overlook smaller lesions among patients with early-stage PDAC [65] as well as occult and unsuspected tumour metastasis. In the latter case though, the staging laparoscopy is used to supplement the limitations of non-invasive imaging techniques [66]. 

The designation of PDAC grading and staging are determined based on American Joint Committee Cancer (AJCC) Staging Manual [67,68] that take both the histopathological grading (G) and TNM scores into consideration [69,70]. The histopathological gradings (G1 to G3) are assigned based on the levels of glandular differentiation and pattern of tumour growth in the neoplastic pancreatic stroma on haematoxylin and eosin-stained tissue sections [69]. On the other hand, the TNM system is an expression of the anatomic extent of the primary tumour (T), presence or absence of regional lymph node metastasis (N), and the presence or absence of distant metastasis (M) while the numerical subsets of the TNM components (T0, T1, T2… M1), indicate the progressive extent of the malignancy. 

Pancreaticoduodenectomy or best known as the Whipple procedure remains the standard surgical treatment of care [71], depending on the location of the PDAC. Unfortunately, since most PDAC cases are clinically presented at a very advanced stage, only less than 20% of patients qualify for surgical resection [72]. Nevertheless, the 5-year survival rate of patients with PDAC who had undergone successful surgical resection was reported to range between 15% and 40% and despite advances in surgical techniques, the rate of tumour recurrence remains as high as 80% [73]. On the other hand, patients diagnosed with locally advanced unresectable PDAC [72] are otherwise subjected to palliative therapy/care (e.g., chemoradiotherapy or stereotactic body radiotherapy) and other appropriate disease management practice [74]. 

Since 1997, gemcitabine, a drug that interferes with DNA synthesis, has been approved by the US Food and Drug Administration (FDA) and accepted as the first-line therapy for the alleviation of PanC symptoms in addition to exhibiting moderate improvement in patient survival rates [75]. Given its favourable effect, combination drug therapies consisting of gemcitabine and other cytotoxic agents such as paclitaxel and docetaxel are now under clinical trials for PDAC [76,77]. Aside from this, FOLFIRINOX, a multidrug containing fluorouracil, leucovorin, irinotecan, and oxaliplatin which was observed to improve the overall survival of PDAC patients [78] is also under clinical trial but intended for the treatment of metastasised PDAC [79]. Although both surgery and drug therapy persist as the standard treatment for PDAC, identification of novel biomarker(s) remains a necessity particularly in assessing the suitability of the selected treatment modalities, thus paving the way for personalised medicine and care of patients.

### 1.2. FDA-Approved Biomarkers for PDAC

The FDA plays an important role in the medical sector in which standards are established for the implementation of biomarkers into clinical practice [80]. These biomarkers could either function as routine diagnostic tests, for evaluation of prognosis or to monitor treatment response and tumour recurrence in patients. In 1981, researchers discovered the over-production of cancer antigen 19-9 (CA 19-9) in patients with PDAC, apart from those with colon carcinoma [81], which then later, was extensively studied for its potential use in the management of PDAC [82]. Currently, CA 19-9 is the most routinely and widely applied biomarker for PDAC that has been approved by the FDA [83,84]. The standard clinical threshold levels of CA 19-9 is at 37 U/mL [82] and patients with increased levels are at high risk of developing PDAC [81]. However, the utility of CA 19-9 in PDAC remains obscure owing to various interpretations of its applications in PDAC in the literature [81]. Although commonly used as a diagnostic marker [82], particularly in combination with imaging modalities such as MRI [85], the clinical utility of CA 19-9 better serves to provide information on prognosis and overall survival [82], monitor treatment responses [86], predict post-operative recurrence and prognosis [87], as well as to predict tumour stages and respectability in PDAC patients [88]. Nevertheless, elevated CA 19-9 levels can also be caused by biliary obstruction, endocrinal, gynaecological, hepatic, pulmonary, and spleen diseases [89] as well as other malignancies (e.g., colon, stomach, lung) [90,91]. Furthermore, individuals with Lewis antigen-negative phenotype (lack of Lewis glycosyltransferase) do not express CA 19-9 [92], thus undermining the use of CA19-9 as a diagnostic biomarker for PDAC in this cohort.

Another FDA-approved tumour marker that has been reported in various studies in the context of PDAC is the use of carcinoembryonic antigen (CEA). The increased production of CEA was initially detected, and subsequently implemented for use as biomarker for colorectal cancer in the 1960s [93]. However, it was later discovered that CEA levels were also elevated (>5 ng/mL) [94] in approximately 30–60% of patients with PanC [95] and significantly associated with poor prognosis and worse overall survival [96]. However, CEA alone is deemed unsuitable for screening for PDAC due to its lower diagnostic accuracy [97], but as a vital supplement to CA19-9 [82,95]. On a brighter note though, CEA can be potentially used as a diagnostic marker for Lewis antigen-negative individuals instead [95].

## 2. Proteomics-Based PDAC Research: Techniques, Samples, and Samples Processing

The advancements in proteomic technology in recent years has enabled an in-depth exploration at cellular and molecular levels for a better understanding of complex diseases such as cancer [98]. The wide range of proteomics tools and technologies allow systematic, comprehensive and/or targeted analyses of structure, function, expression, interactions, and modifications of proteins [99]. Recently, non-gel-based separation and detection proteomics techniques, particularly hyphenated technology, which are typically represented by an online combination of a separation technique and one or more spectroscopic detection techniques has fast gained popularity [100]. Amongst the hyphenated technologies, liquid chromatography tandem mass spectrometry (LC-MS/MS) has emerged as the most preferred methodology [101] for proteomics-based (cancer) research [102,103,104].

Further, quantitative proteomics has recently developed in its ability to generate reasonably accurate quantitation of the expression of proteins, especially in cancer research [105]. The coupling of mass spectrometry that uses ionisation techniques, paired with mass analysers with quantitative labelling strategies was reported to improve the detection and quantification of proteins [106]. Moreover, MS-based targeted proteomics methods such as multiple reaction monitoring (MRM) and parallel reaction monitoring (PRM) are now rising as promising tools for the validation of identified proteins in biomedical applications [107].

Proteomics technologies were likewise widely applied in the investigation of PDAC in pursuit of the identification of potential PDAC-associated proteins intended for diagnosis and staging, assessment of tumour resectability, prognosis and predicting responses to treatment in patients [108]. For this, various clinical specimens including pancreatic juice [109], tumour tissues [110] and cyst fluid [111], plasma or serum [112] and urine [113] have been previously extensively used.

In addition to this, PanC cell lines are a useful source and/or model for biomarker investigations. This is because cell lines can be easily obtained and allow the analysis of secreted proteins by means of culturing and harvesting the cells [92]. Different PanC cell lines (e.g., Capan-2, PANC-1) exhibit distinct phenotypes and genotypes that represent the different subclasses of PDAC (e.g., classical, quasi-mesenchymal, respectively) which when their proteins are profiled, may provide insights on the differential expressions of specific proteins associated with tumour growth [114], metastasis [115] as well as response to therapy [116]. For example, by comparing a primary cell line, BxPC-3, with a metastatic cell line, AsPC-1, researchers were able to detect the differentially expressed proteins involved in the metastasis of PDAC that may serve as potential biomarker(s), once validated [117].

Aside from this, the stromal compartment, which constitutes 80–85% of the tumour [118], consisting of the extracellular matrix [119] infiltrated by cancer-associated fibroblasts (CAF) [120], inflammatory cells [121], and immune cells (e.g., lymphocytes and macrophages) [122] is another choice of sample for biomarker investigations. Researchers were able to identify differentially regulated proteins in the stroma of PDAC by co-culturing the PanC cell lines together with the components of stroma such as CAF [123]. For example, the differentially regulated proteins expressed in the stromal compartment such as hemopexin were previously reported to regulate the progression of tumours, thus revealing association with lymph-node metastasis resulting in poor prognosis in patients [124]. Similarly in another study, Tao et al. reported that PDAC has an extensive stroma and abundant extracellular matrix that lacks vascularisation, resulting in hypoxia within the pancreatic tumour environment, thus revealing candidate biomarkers that may be useful in monitoring the treatment response in PDAC patients [125].

On the other hand, exosomes are extracellular vesicles produced by cells that are involved in cell communication [126]. The isolation and identification of cancer-specific proteins derived from exosomes may be developed into potential candidate biomarkers since the altered protein profiles of exosomes correlate with the pathogenesis of cancer [127]. For example, Melo et al. reported glypican-1 (GPC1), a cell surface proteoglycan that is specifically enriched in exosomes derived from cancer cells in PDAC patients by using MS techniques [128]. According to their research, GPC1 may serve as a potential biomarker for early stage PDAC. This shows that the analysis of exosomes can also be a useful target for potential clinical utility.

Glycosylation, a type of post-translational modification of proteins, is a hallmark of cancer [129]. As such, targeting the aberrations occurring during glycosylation either in terms of the levels of expressions of proteins or the glycans (glycome; e.g., truncated *O*-glycans, increased branching and fucosylation of *N*-glycans, upregulation of specific proteoglycans and galectins, and increased *O*-GlcNAcylation [130]) may be beneficial for the identification of diagnostic and/or therapeutic targets [131] as well as for aiding in treatment decisions for PDAC [132]. Along this line, mucin, a highly glycosylated protein, could also be a potential biomarker [133]. The levels of expression of mucin were previously reported to be altered in the course of development and progression of PDAC [134]. According to Wang et al., PDAC tissues expressed high levels of MUC1, MUC4, MUC5AC, MUC5B, MUC6, MUC13 and MUC16, all of which are involved in promoting the aggressive phenotype of this disease [134]. These findings further highlight the roles of aberrant protein glycosylation in the progression of PDAC.

The most critical step in a proteomics workflow is the optimal preparation and/or processing of samples, which in turn, ascertain the degree of sensitivity for detection of proteins in downstream application [135]. The complexity of the samples due to high biological variations and/or post-translational modifications remains a major limitation in proteomics study. Furthermore, samples such as serum and/or plasma contain significant amounts of proteins of higher abundance (e.g., albumin, immunoglobulins) that are involved in various yet important biological and physiological functionality, hence may not be useful as potential markers [136]. In turn, due to their predominant presence, these proteins are infamously known for masking away other proteins, particularly low molecular weight proteins that are present in lower abundance, but may have biomarker potential (e.g., small secreted proteins or peptide hormones) [137]. The complexity of serum or plasma is usually reduced via depletion of high-abundance proteins [138] or using sub-proteome-specific enrichment method [139]. While depletion workflows help to improve the detectability of low-abundance proteins, enrichment methods are focused on increasing the sensitivity of targeted sub-proteomes for detection [140]. On this note, Hashim et al. reviewed the use of lectins to enrich glycosylated proteins which are differentially expressed in cancer, as they might not be detected using conventional methods due to their low abundance [139]. Apart from this, immunoprecipitation is another method commonly used to enrich target proteins by using antibodies that bind to specific antigens resulting in immune complexes that are captured on solid phase support such as chromatography resin and magnetic beads [141].

## 3. Biomarker Investigations

An ideal biomarker should be able to reliably characterise a particular disease status and/or outcome, hence, shall be disease and/or condition specific (100% specificity) and highly sensitive (100% sensitivity) (e.g., everyone with and without cancer shall test positive and negative, respectively, for the biomarker) [142]. On the same note, a reliable biomarker should also meet other criteria including being robust and well-validated thus allowing effective use and implementation in clinical routines [143]. With this, the subsequent section discusses the protein signatures identified as potential biomarkers in assorted biological matrices for various clinical applications of PDAC using proteomics technologies in recent years.

### 3.1. Biomarkers for Early Detection and/or Diagnosis of PDAC

Early detection and reliable diagnosis for PDAC is paramount. This is because it would not only result in the identification of eligible patients for surgical resection but consequently, improve the overall survival of patients [17]. In line with this, Guo et al. investigated concanavalin A enriched *N*-glycosylated proteins from the pre-and post-operative serum of patients with PDAC using offline liquid chromatography (LC) coupled to a matrix-assisted laser desorption/ionisation-time of flight mass spectrometry (MALDI-ToF-MS) [144]. Together, the investigation led to the identification of dysbindin to be significantly correlated with the size and differentiation of the tumour. The overexpression of dysbindin in pre-operative sera was thought to promote the phosphorylation of PI3/Akt signalling pathway [145], which in turn stimulates the proliferation of the cancer cells. In addition, the subsequent validation set assessed via ELISA further confirmed the high levels of specificity (73.9%) and sensitivity (82.3%) of dysbindin for PDAC, and also demonstrated improved diagnostic performance compared to CA 19-9. Interestingly, a study by Fang et al. reported an overexpression of dysbindin in patients with PDAC to correlate with the size of tumour and histological differentiation, thus suggesting its role-play in the prognostic measure of patients [145]. Likewise, another study by Zhu et al. showed that dysbindin promotes metastasis of PDAC through the activation of NF-κB/MDM2 signalling pathway, further indicating that this protein additionally assumes the role of prognostic predictor [146]. Based on these studies, it can be postulated that dysbindin plays more than one role in the pathogenesis of PDAC, hence the use of this protein for (any) clinical utility must first be carefully clarified.

In a similar source of samples and enrichment methods but using tandem mass tags (TMT) labelling and LC-MS/MS, Sogawa et al. had otherwise reported increased expression of glycosylated 4b-binding protein α-chain (C4BPA) and polymeric immunoglobulin receptor (PIGR) in the pre-operative serum of PDAC patients than in post-operative patients [147]. However, upon validation using ELISA, only the elevated levels of C4BPA which was due to the host immune responses against the tumour, remained consistently significantly high in patients with PDAC when compared to those with pancreatitis and healthy controls. To further assess the ability of C4BPA as a specific biomarker for PDAC, the researchers also compared the serum levels of C4BPA in comparison with CA 19-9 in other types of gastroenterological malignancies (e.g., biliary tract cancer). Interestingly, they found that the AUC values of C4BPA were much higher than CA 19-9, suggesting this protein could indeed be a specific biomarker for PDAC. However, studies have also shown that this protein is additionally highly expressed in other cancers such as ovarian cancer [148] and breast cancer [149].

Nevertheless, four years later, Sogawa and colleagues then applied ELISA by using lens culinaris agglutinin (LCA)-lectin that binds specifically to fucose, to measure the levels of fucosylated (Fuc-) C4BPA in the serum of pre-operative PDAC patients, chronic pancreatitis patients and healthy controls [150]. In various validation sets with different sample groups, Fuc-C4BPA was found to be upregulated in pre-operative PDAC patients. Moreover, in comparison to total C4BPA, CA 19-9 and CEA, Fuc-C4BPA showed higher AUC values for discriminating PDAC patients from other groups of subjects, thus counter-suggesting it as a potential diagnostic biomarker instead. Furthermore, the upregulation of Fuc-C4BPA is not reported in other types of cancers to date. Intriguingly, the researchers also discovered that Fuc-C4BPA was able to predict lymph node metastasis. This is particularly useful as the prediction of metastasis would aid in the treatment decision for PDAC patients [151].

Earlier, Kim et al. had reprogrammed PDAC cells into induced pluripotent stem cell-like lines to study the development of the disease by isolating epithelial cells and then inducing reprogramming factors such as *Oct4*, *Sox2*, *Klf4*, and *c-Myc* [152]. The normal-phase (NP)-LC-MS/MS analysis of the iPSC-like lines revealed 43 differentially regulated proteins that were associated with transforming growth factor-β and integrin signalling involved in the development of PDAC. Four years later, the same group of researchers focused on just three proteins namely, matrix metallopeptidase 2, matrix metallopeptidase 10 and thrombospondin-2 (THBS2) for validation using ELISA [153]. Nevertheless, the results only substantiated the previous works by Kim et al. for elevated levels of plasma THBS2 in patients with PDAC compared to patients with benign pancreatic disease and healthy controls [152]. Further, when THBS2 was assessed for accuracy in combination with CA 19-9, together they yielded a specificity of 98% and sensitivity of 87% for the diagnosis of PDAC. However, in recent research, Le Large et al. found that the plasma THBS2 levels of PDAC and distal cholangiocarcinoma (dCCA) patients were significantly higher than in patients with benign pancreas diseases and healthy controls [154]. Although the clinical symptoms of PDAC and dCCA are relatively similar, these diseases have distinct entities and require specific biomarkers for discrimination [155]. In view of this, the specificity of THBS2 as a biomarker for PDAC remains to be determined.

Years ago, Nakamura et al. identified 260 genes that were upregulated in PDAC [156]. Almost two decades later, as a follow-up study, Yoneyama et al. had, in turn, explored the combinatory potential of antibody- (reverse-phase protein array (RPPA)) and LC-MS/MS-based proteomics in their quest to identify new PDAC diagnostic biomarkers, by focusing on only 130 encoded proteins having known functions and available commercial antibodies [157]. Based on the RPPA-based biomarker screening, the researchers then chose only 23 proteins for validation by MRM-MS. Of these, significantly different reciprocal levels of insulin-like growth factor-binding protein 2 and 3 (IGFBP2 and IGFBP3, respectively) were observed in the plasma of patients with early stage PDAC compared to control subjects. In agreement with this finding, Baxter had previously explained that these proteins consisting of Arg-Gly-Asp motif bind to integrins, thus resulting in the stimulation of cancer cell proliferation [158]. Moreover, it was revealed that the combinatory assessment of IGFBP2 and IGFBP3 with CA 19-9 effectively discriminates early-stage PDAC patients from healthy controls by recording an AUC value of 0.9 [157]. Contradictorily though, IGFBP2 too was previously reported to induce epithelial to mesenchymal transition, which is involved in metastasis of PDAC, suggesting its prognostic role in PDAC patients [159].

Previous research has indeed reported that a single biomarker, such as CA 19-9 alone is unable to provide a reliable diagnosis that is sensitive and specific for PDAC [160]. As deduced based on a few studies below, biomarker panels that combine a few markers appear effective and enhance the accuracy of PDAC diagnosis. For this, Jisook Park et al. [161] and Jiyoung Park et al. [162] had also demonstrated improved cancer discerning capabilities of the identified respective panels of proteins when used in combination with CA 19-9, thus concurring with the previous hypothesis of having multi-markers for complex diseases such as cancer rather than just a single biomarker [163]. Here, Jisook Park et al. first identified the promising candidate biomarkers through shotgun proteomics and pathway-based gene expression meta-analysis, then further validated the selected nine protein candidates via stable isotope dilution (SID)-MRM-MS and immunohistochemistry [161]. Based on the results, apolipoprotein A-IV (APOA4), apolipoprotein C-III, IGFBP2 and tissue inhibitor of metalloproteinase 1 (TIMP1) were found significantly altered in the serum of PDAC patients (stage I–IV) compared to those with pancreatitis as well as healthy controls. These are acute-phase proteins and are strongly associated with cancer and its development [164] and hence, has been previously suggested as potential biomarkers [165]. For instance, acute phase proteins including α1-antitrypsin, α1-antichymotrypsin (ACT), complement factor B (CFB) and leucine-rich glycoprotein (LRG) proteins were previously reported to be enhanced in PanC, while upregulated levels of ACT, CFB and clusterin as well as decreased levels of kininogen in patients with breast cancer [165]. By comparing the diagnostic performances of these four different proteins in combination with CA 19-9, the researchers then proceeded to generate a biomarker panel consisting of APOA4, TIMP1 and CA 19-9 that showed better performance in distinguishing early stage PDAC (stage I and II) from those with pancreatitis (90% specificity and 85.5% sensitivity).

Following an extensive database and literature search and review of over 1000 candidate markers, Jiyoung Park et al. refined and selected two candidate proteins consisting of leucine-rich alpha-2 glycoprotein (LRG1) and transthyretin (TTR) in combination with CA19-9 for validation using MRM-MS on more than 1000 plasma samples [162]. The performances of the panel were evaluated in various conditions: PDAC stage I and II vs. healthy controls, PDAC vs. benign pancreatic disease and other cancers individually. Overall, it was observed that the biomarker panel had a sensitivity of 82.5% and a specificity of 92.1%. To further establish this biomarker panel, the researchers then developed an automated multi-marker ELISA kit using the three proteins for the diagnosis of PDAC and observed enhanced levels of specificity at 90.69% and sensitivity at 92.05%. Nonetheless, the inclusion of TTR as a biomarker is rather conflicting considering another report which has demonstrated higher levels of TTR in the sera of patients with PDAC compared to controls but using 2D-DIGE, MALDI-ToF-MS and validation via ELISA [166].

Serological proteome analysis (SERPA), also known as 2D western blot analysis, is a technique used to identify tumour antigens by first fractionating the cell lysates with 2D gels followed by transfer of the proteins onto a membrane and probing with serum [167]. By using SERPA, 18 immunoreactive antigens were identified in serum via 2-DE and MALDI-ToF-MS. These include ATP synthase, glyceraldehyde 3-phosphate-dehydrogenase (GAPDH), laminin, phosphoglycerate mutase B (PGAM-B), Rho GDP-dissociation inhibitor II (RhoGDI2), septin, superoxide dismutase (SOD) and tubulin β8 channel, all of which were found strongly associated with the pathogenesis of PDAC [168]. Here, the researchers discussed the roles of different types of immunoreactive proteins such as cytoskeletal proteins (e.g., laminin, septin and tubulin β) and metabolic reprogramming-associated proteins (e.g., GAPDH, PGAM-B, RhoGDI2 and SOD) in cancer. Nonetheless, some of these proteins are additionally regarded as general proteins that take part in key processes in the cell, therefore, the real mechanism with which these proteins are associated with the diagnosis of PDAC remains unknown.

To date, most of the biomarker studies on PDAC are typically based on serum/plasma and tissue analysis. Only quite recently, a few publications have highlighted the potential of urine as an interesting biological sample for biomarker investigations in PDAC have emerged. For instance, Radon et al. analysed the proteome of urine samples obtained from PDAC and chronic pancreatitis patients, as well as healthy controls using NPLC-MS/MS [169]. They identified a candidate biomarker panel consisting of lymphatic vessel endothelial hyaluronan receptor 1 (LYVE-1), regenerating family member 1 alpha (REG1A) and trefoil factor 1 (TFF1). Following this, since there were several studies [170,171,172,173] that have proposed REG1A as well as REG1B from various biological samples such as serum, urine, tissues and pancreatic ductal fluid as candidate biomarkers, five years later, the same group of researchers replaced REG1A with REG1B for the validation of the biomarker panel using ELISA [113]. They then compared the performance of this newer protein panel with the previous study and found that the urinary REG1B levels (AUC value: 0.93) outperformed REG1A (AUC value: 0.90) in discriminating early stage PDAC (stage I and II) from the healthy controls as well as chronic pancreatitis patients. Concurring with this finding, Li et al. also showed an increased expression of REG1A but in tissues of PanIN lesions as they progress to PDAC, while the expression of REG1B remained elevated only in the early stage of PanIN lesions, thus highlighting REG1B as a better choice for use as a diagnostic biomarker [170]. In addition, Li et al. reported that although the serum levels of both REG1A and REG1B were significantly higher in PDAC patients compared to healthy controls, there was also an insignificant elevation of these proteins in chronic pancreatitis patients as compared to healthy controls [170]. When both studies [113,170] were compared in this context, it highly indicated a differential expression of these proteins in the different types of biological samples. On a different note, Li et al. reported a prognostic behaviour of these proteins as such that the expression of these proteins was seen to gradually reduce as the tumour progresses from well differentiated to poorly differentiated [170]. This finding certainly contradicts the study of Debernadi et al. on the potential of REG proteins as diagnostic biomarkers for PDAC [113].

Table 2 summarises the recently identified biomarkers for the early detection and/or diagnosis of PDAC.

### 3.2. Biomarkers for Determining Prognosis of PDAC

The prediction of prognosis is important in determining the likely health outcome of cancer patients (e.g., overall survival, disease recurrence). For this, Kuwae and colleagues attempted to identify a biomarker with prognostic potential by analysing the proteomes of tumours and adjacently located non-tumour pancreatic tissues of the same patient using iTRAQ labelling and NPLC-MS/MS [175]. In this study, the researchers utilised Zwittergent-based buffer for the extraction of proteins from the formalin-fixation and paraffin-embedded tumour tissues for LC-MS/MS analysis. In line with this, a study by Shen et al. comparing the different extraction buffers for downstream proteomic analysis of tissue samples deduced that Zwittergent was the most effective and efficient for protein extraction in these sample types [176]. Here, the elevated levels of paraneoplastic Ma antigen–like 1 (PNMAL1) was found in tumour tissues but only in trace amounts in the adjacent non-tumour tissues. Furthermore, immunohistochemistry analyses revealed that positive expression of PNMAL1 was significantly correlated with better overall survival compared to those patients with negative expression. In contrast to this study though, Jiang et al. reported a decreased viability of the PanC cell lines following *PNMAL1* silencing, thus indicating that PNMAL1 is an anti-apoptotic factor that promotes the survival of cancer cells [177]. The discrepancies observed may have resulted due to the employment of different methods of analysis (protein vs. gene expression) as well as the different types of samples used (tumour tissues vs. cell lines) in the respective studies. At the same time, these studies have also indicated that the mechanism of function of PNMAL1 in association with PDAC is not fully understood. In a different study though, this protein was reported to exert a pro-apoptotic function in neurons and its elevated expression was postulated to contribute towards neurodegenerative disorders [178].

Over the years, the overexpression of survivin in the context of PDAC has been widely studied [179,180,181,182]. Survivin is a member of apoptotic inhibitor protein that is reported to inhibit apoptosis in PDAC cells, hence, inversely correlated with the prognosis of PDAC as well as with higher rates of recurrence [183]. Using tissue microarray and immunohistochemistry, Zhou et al. had also studied the expression of survivin in the PDAC tumour and adjacently located non-tumour tissues obtained from the same patient [184]. Higher expression of survivin in the tumour tissues of patients further corroborates the previous reports for its strong association with poor prognosis of the disease via Kaplan–Meier survival analysis.

On the other hand, Bauden et al. conducted NPLC-MS/MS analysis on tumour tissues obtained from PDAC patients and normal pancreas head biopsy tissues from organ donors [185]. In this study, the team had identified histone variant H1.3 to be differently expressed and was further validated via immunohistochemistry analysis. The analysis demonstrated a decreased survival for patients with positive H1.3 expression, suggesting that this protein may serve as a prognostic biomarker for PDAC. In general, the alterations in the epigenetic processes which involve the modifications of histone variants are known to modify cell cycle progression, thus resulting in the development and progression of cancer (Ferraro, 2016). Since histone variant is also found to participate in the epigenetic regulation of PDAC which in turn contributes to the aggressiveness of the disease, hence, profiling of histone variants may prove as a useful method for identifying biomarkers of PDAC.

In another study, alpha-1-acid glycoprotein 1 (AGP1) was found upregulated in PDAC tissues compared to normal pancreatic tissue obtained from patients with benign pancreatic disease such as serous cystadenoma, mucinous cystadenoma and pancreatic pseudotumor via NPLC-MS/MS and later, verified by PRM [186]. Additional analysis using tissue microarray and immunohistochemistry (and, statistical analysis) also revealed that this protein significantly correlates with worse overall survival. Pathway analysis, on the other hand, demonstrated that this protein is prominently involved in the signalling cascade related to PDAC cell proliferation, migration, and invasion including MAPK, p53 and YY1 signalling thus indicating its potential as a prognostic biomarker for PDAC. At the same time, this study has considered the use of this protein as part of a biomarker panel for the early detection of PDAC as well [186]. The aberrant expression of AGP1 in PDAC has been discussed in other previous studies [187,188,189]. For example, Balmaña et al. utilised several analytical techniques (zwitterionic hydrophilic interaction capillary liquid chromatography electrospray ionisation-MS coupled with capillary zone electrophoresis and enzyme-linked lectin assay) to identify AGP1 glycoforms that are associated with PDAC [187]. In this study, *α*1-3 fucosylated glycoforms of AGP1 was observed elevated in the serum of PDAC patients compared to chronic pancreatitis patients and healthy controls. On the other hand, since this protein is associated with the signalling pathway in cancer cells, there is a high possibility for this protein to be upregulated in other cancers as well. Confirming this statement, Zhang et al. and Ayyub et al. reported an increase in the expression of this protein in laryngeal and lung cancers, respectively [190,191].

Quite different from the above studies, Kim et al. previously identified fibrinogen as a potential biomarker in the serum of PDAC patients as compared to healthy controls using MALDI-ToF/MS [192]. Five years later, they then validated this protein in PDAC patients, diabetic patients and healthy controls using ELISA and found that the serum fibrinogen levels were higher in PDAC patients compared to the healthy control group [193]. However, the analyses showed that the sensitivity and specificity of fibrinogen in discriminating PDAC and diabetic patients ranged between 67.4% and 83.6%, respectively, thus dismissing fibrinogen as a potential diagnostic biomarker for PDAC. Nevertheless, the same protein was found present at higher levels in patients with distant metastasis compared to those without when assessed for its prognostic values instead, thus indicating its correlation with poor prognosis of PDAC [193]. On the same note though, fibrinogen is an acute-phase protein that plays a common role in blood clotting and inflammatory response [194]. Hence, although its levels were indeed found to increase in advanced tumour stages, the specificity of this protein in PDAC remains to be determined due to the common role of this protein in other cancers [195,196,197,198] as well as other inflammatory conditions [199,200].

Table 3 summarises the recently identified biomarkers for the prognosis of PDAC.

### 3.3. Biomarkers for Monitoring Treatment Response and Predicting Tumour Recurrence in PDAC

Although survival rates of patients with PDAC have improved to a certain extent (1-year survival rate of 18%) using gemcitabine [201], not all patients have benefited equally from this therapeutic regimen. This is due to the presence of extensive and dense stroma of PDAC attributing to the inability of the drug to penetrate, thus contributing to chemoresistance of gemcitabine [202]. Hence, biomarkers are needed for monitoring the response of patients post-treatment [203]. However, based on the review of recent research on PDAC, most biomarker studies that have been published using proteomics approaches were intended for diagnostic and prognostic applications. Conversely, biomarker studies focusing on monitoring treatment response among patients with PDAC appears to predominantly prefer genomic or transcriptomic approaches [26,204,205,206,207]. Hence, we report here only those studies that fell into the scope of this present review.

For example, Peng et al. compared the proteome profile of plasma obtained from PDAC patients who responded positively to chemotherapy and had longer survival (>12 months) with patients who responded poorly to treatment and had shorter survival (<12 months) to identify proteins that were differentially expressed between the two groups of patients via RPLC-MS/MS [208]. They discovered three proteins, including vitamin-K dependent protein Z (PZ), sex hormone-binding globulin and von Willebrand factor (VWF), which together with CA 19-9, provided better results in distinguishing patients that would benefit from chemotherapy. In this study, PZ, a protein that is involved in regulating blood coagulation [209] was observed to be highly abundant in positive treatment response patients. Although the mechanism of PZ in PDAC has still not been explored, a previous study on gastric cancer showed that decreased levels of PZ corresponded with advanced disease stage, suggesting that the varying levels of this protein may indicate a tumour stage [210]. However, like few other proteins discussed earlier in the present review, the validity of PZ as a biomarker for PDAC is still pre-mature since the biological significance of this protein has yet been elucidated.

Besides biomarkers for monitoring response of PDAC patients towards treatment, there are also studies that have been conducted to identify potential biomarkers for predicting recurrence of PDAC after the Whipple procedure for guiding and administering personalised treatment(s) in patients, if identified and validated.

Previously, Hu and group [211] identified galectin 4 as one of the proteins that are upregulated in the tissues of PDAC patients with longer survival (>45 months) through RPLC-MS/MS and later verified the data via PRM. Two years later, the same group of researchers conducted an immunohistochemical study to identify the significance of this particular protein in predicting the recurrence of PDAC among patients who had undergone surgical resection [212]. Here, they found galectin 4 to be significantly linked with disease recurrence within the first year of surgery and survival of patients after a year. Interestingly, another study by Kuhlmann et al. [213] similarly discovered galectin 4 as a biomarker candidate to monitor treatment response and tumour recurrence specifically for the exocrine-like subtype of PDAC. Under normal physiological conditions, galectin 4 plays various biological and functional roles including participating in apical trafficking, lipid raft stabilisation, aiding in the healing of intestinal wounds and promoting growth of axons in neurons [214]. In inflammatory and/or cancer conditions, on the other hand, this protein has been reported to exacerbate intestinal inflammation and promote tumour progression [214]. Further, galectin 4 additionally appears to have a conflicting pattern of expression in other cancers. For example, this protein was overexpressed in the serum and tissues of patients with cervical cancer [215] and lung adenocarcinoma [216]. While, at the same time, it was also found downregulated in expression in the tissues of patients with metastatic hepatocellular carcinoma [217] and colorectal cancer [218]. Hence, further studies are needed to investigate whether galectin-4 should be implemented for use in clinical settings.

Table 4 summarises the recently identified biomarkers for monitoring treatment response and tumour recurrence of PDAC.

Of note, apart from the above-mentioned biomarker studies on PDAC, there is also other literature on the less common types of malignancies such as pancreatic neuroendocrine neoplasm [219], pancreatic neuroendocrine tumour [220] and insulinoma [221] using various proteomics approaches.

## 4. Challenges and Future Directions

Presently, from the perspective of diagnosis for PDAC, identifying an accurate and low-cost screening test for the early detection of PDAC remains a challenge but is of utmost importance. However, due to the low incidence of this disease, conducting screening tests on a population of a larger scale seems impossible. Another major challenge faced by healthcare professionals is the ability to distinguish early stage PDAC from other benign pancreatic conditions such as chronic pancreatitis [222]. In view of this, functional markers that could indicate the progression of PDAC such as stromal changes, microvascular density and tumour metabolism [223] in addition to studies that focus on other modifiable risk factors for PDAC such as diabetes mellitus [224] and obesity [225] are currently being studied. Furthermore, most treatment options that typically involve chemotherapy were found to have no improvement on patient life expectancy [226].

On the other hand, from the angle of study design, samples for biomarker research are usually obtained and validated in a case vs. control type of study and hence, the low prevalence of said disease in a particular population is often ignored. Taking the recent past research findings and its output into consideration, a longitudinal type of study would be deemed a better design/model for such undertakings. Secondly, the sample size in need of consideration. Most proteomics studies employ only a small sample size which in essence might not represent the actual prevalence or presentation of a particular disease in a population. A large cohort of samples is highly necessitated to establish ‘real’ biomarkers in a clinical setting. Thirdly, the choice of (biological) samples requires further refinement. For example, the heterogeneity of cancerous tissues is often not considered especially when tissue samples are used for the experiment [227]. This is because tissues are often homogenized prior to protein extraction for use for proteomic studies. On the other hand, cell lines may not accurately represent the primary cells of a particular cancer/tumour [228]. This is because cell lines are generally genetically manipulated and thus may have resulted in the alteration in their phenotype that might be distinct from the phenotype of the actual tumour [229].

Large amounts of money have been invested in the acquisition of proteomics technologies worldwide, with the hope of exploiting these advanced technologies for identifying highly specific and sensitive biomarkers with validated clinical outcomes. In spite of this, unfortunately, to date, only very few biomarkers have shown any significant clinical impact, if at all [230]. At the same time, implementation of such advanced (proteomic) techniques (e.g., MS) to measure the biomarkers (albeit validated) in a common clinical setting may not be practical and economical in terms of cost of equipment and (skilled) labour requirements.

Further, in most of the studies, the roles/mechanisms of the identified proteins in the pathogenesis of PDAC (e.g., why, and how the expression of proteins are correlated with the extent that they are secreted in normal or diseased conditions) remains unclear due to the singularity of techniques used thus limiting the interpretation of the significance of the study. This by no means only applies to proteomics but other fields of research as well (e.g., genomic [231,232], epigenomic—DNA methylation [233], single-cell transcriptomic [234] and metabolomics approaches [50,235]). One way to solve this is via the integration of multi-omics technologies that combine various approaches such as genomics, epigenomics, transcriptomics and metabolomics, together with proteomics. Such a strategy is highly anticipated to translate the results of exploratory research into routine clinical practice, be it for either early detection and diagnosis, prognosis prediction or even to monitor treatment response and/or tumour recurrence.

Adjunct to the application of proteomics in biomarker investigations for PDAC, various other newer technologies are under development. For example, the nanoparticle-enabled blood test [236], incorporation of artificial intelligence into scientific discovery [237], scent test using *Caenorhabditis elegans* [238], in addition to the possible use of volatile organic compounds [239] for the management of patients with PDAC.

## 5. Conclusions

PDAC is the most prevalent disease of the pancreas, accounting for approximately 90% of all pancreatic malignancies. This disease has a poor prognosis due to the lack of early detection methods and is typically diagnosed at a late stage. Developing reliable, specific and sensitive biomarkers is of great importance to guide in the diagnosis of PDAC at an early stage and ascertain the prognosis of patients in order to serve as a guide in the timely and effective treatment of the disease [230]. The discovery of differently regulated yet unique protein signatures for various clinical utilities of PDAC via a proteomics approach provides deeper insights into cell functions, pathways and biological processes that are involved in the development and progression of the disease. The ever-evolving proteomics technologies have enabled researchers to grasp a basic understanding of the mechanisms of the disease, and for the identification of potential proteomics-based markers for PDAC, albeit with many challenges. Nevertheless, most of them only appear to demonstrate moderate sensitivity and/or specificity and are far from being considered for application in clinical settings. Hence, future biomarker investigation studies should essentially include several prerequisites such as the inclusion of an adequate number of clinically representative samples/populations, and improved yet appropriate study designs. Further, the incorporation of robust multi-omics (combination of genomics, epigenomics, transcriptomics and metabolomics with proteomics) and/or other newer technologies is hoped upon to lead to the discovery of reliable diagnostic, prognostic, and biomarkers to monitor treatment responses that could be implemented into clinical practice in the near future.

## Figures and Tables

**Figure 1 ijms-23-02093-f001:**
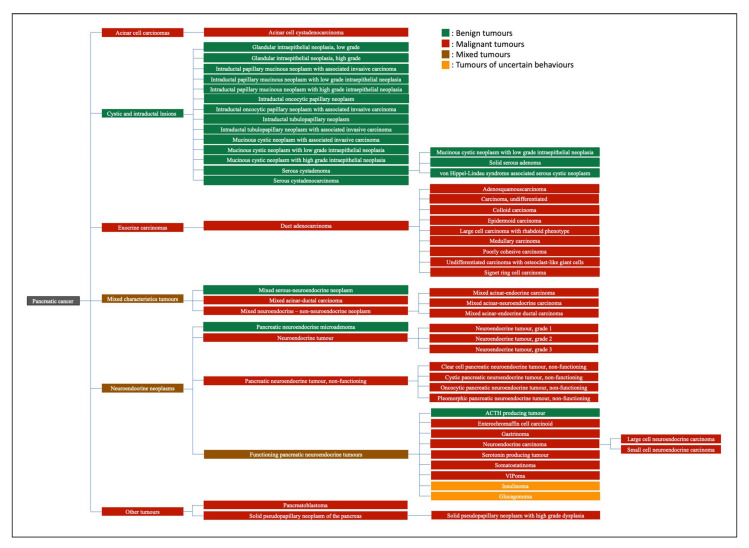
Classification of pancreatic cancer.

**Table 1 ijms-23-02093-t001:** Selected recently identified biomarkers for PanC.

Target *	Name	Clinical Utility	References
DNAs	*K-ras* mutation	Diagnosis	[23]
	Methylated *ADAMTS1* and *BNC1*	Early diagnosis	[24]
	*TP53* mutation	Prognosis	[25]
	Mutations of *BRCA2, EGFR, ERBB2* and *KDR*	Monitoring treatment response	[26]
	Peritoneal lavage tumour DNA	Prognosis/Monitoring tumour recurrence	[27]
mRNAs	*WASF2* mRNA	Early diagnosis	[28]
	*EVL* mRNA	Prognosis	[29]
	*FAM64A* mRNA	Prognosis	[30]
MicroRNAs (miR) [31] **	miR-181c miR-210	Diagnosis	[32]
	miR-10b miR-155 miR-216	Prognosis	[33]
	miR-196a	Prognosis	[34]
	miR-21	Diagnosis/Prognosis/Monitoring treatment response	[32,35,36]
	miR-155	Monitoring treatment response	[37]
	miR-142-5p miR-506 miR-509-5p miR-1243	Monitoring treatment response	[36]
	miR-451a	Prognosis/Monitoring tumour recurrence	[38]
Long noncoding RNAs	*SNHG15*	Early diagnosis	[39]
	*HOTAIR* *MALAT-1*	Prognosis	[40]
	*LINC00460*	Prognosis	[41]
	*PVT1*	Monitoring treatment response	[42]
Circulating tumour cells		Diagnosis	[43]
		Prognosis	[44]
	Vimentin (surface marker)	Monitoring treatment response	[45]
		Monitoring tumour recurrence	[46]
Metabolites	Panel of acetylspermidine, diacetylspermine, indole-derivative and two lysophosphatidylcholines	Early diagnosis	[47]
	Polyamines	Diagnosis	[48]
	Ethanolamine	Prognosis	[49]
	Lactic acid L-Pyroglutamic acid	Monitoring treatment response	[50]
Carbohydrates (glycan)	Alpha-2,6-linked sialylation and fucosylation of tri- and tetra-antennary *N*-glycans	Diagnosis	[51]
	*N*-glycan branching: alpha-1,6-mannosylglycoprotein 6-beta-*N*-acetylglucosaminyltransferase A	Early diagnosis	[52]
	*β*1,3-*N*-acetylglucosaminyltransferase 6	Prognosis	[53]
	Hyaluronan	Monitoring treatment response	[54]

*ADAMTS1*—A disintegrin and metalloproteinase with thrombospondin motifs 1; *BNC1*—zinc finger protein basonuclin-1; *BRCA2*—Breast cancer susceptibility gene-2; *EGFR*—Epidermal growth factor receptor; *ERBB2*—Erb-b2 receptor tyrosine kinase 2; *EVL*—Ena/VASP-like; *FAM64*—Family with sequence similarity 64 member A; *HOTAIR*—HOX transcript antisense RNA; *KDR*—Kinase insert domain receptor; *KRAS*—Kirsten rat sarcoma viral oncogene homolog; *LINC00460*—Long intergenic non-protein coding RNA 460; *LDLRAD3*—Low density lipoprotein receptor class A domain containing 3; *MALAT-1*—Metastasis associated lung adenocarcinoma transcript 1; *PVT1*—Plasmacytoma variant translocation 1; *RNU2-1*—RNA U2 small nuclear 1; *SNHG15*—Small nucleolar RNA host gene 15; *WASF*-2—Wiskott–Aldrich syndrome protein family member 2. * Recently identified protein-based biomarkers for PDAC will be discussed in the subsequent section of this review. ** This has been previously extensively reviewed by Tesfaye et al. (2019).

**Table 2 ijms-23-02093-t002:** Recently identified diagnostic biomarkers for PDAC using proteomics techniques.

Name	Sample	Proteomics Techniques	Validation			References
			Method	Sensitivity *	Specificity *	
C4BPA	Serum	TMT labelling & LC-MS/MS	ELISA	67%	95%	[147]
Dysbindin	Serum	RPLC & MALDI MS	ELISA	82%	85%	[144]
Panel of APOA1, APOE, APOL1, ITIH3 in combination with CA 19-9	Tissues	iTRAQ labelling & LC-MS/MS	SID-MRM-MS	95%	94%	[174]
Panel of APOA4, TIMP-1 in combination with CA 19-9	Serum	MRM-MS	IHC	86%	90%	[161]
Panel of IGFBP2, IGFBP3 in combination with CA19-9	Plasma	RPPA & LC-MS/MS	MRM-MS	Not reported	Not reported	[157]
Panel of LRG1, TTR in combination with CA19-9	Plasma	Database and literature search	Yes: MRM-MS & ELISA	83%	92%	[162]
Panel of LYVE-1, REG1B and TFF1	Urine		Yes: ELISA	>85%	>85%	[113]
THBS2 and CA 19-9	Plasma	LC-MS/MS	Yes: ELISA	87%	98%	[153]

2DICAL LC-MS/MS—2-Dimensional image converted analysis of liquid chromatography and mass spectrometry; 2-DE—Two-dimensional gel electrophoresis; 2-DIGE—Two-dimensional difference gel electrophoresis; A1BG—Alpha-1B-glycoprotein precursor; ANXA1—Annexin A1; APOA4—Apolipoprotein A-IV; C4BPA—C4b-binding protein α-chain; CA 19-9—Cancer antigen 19-9; CXCL7—CXC chemokine ligand 7; ELISA—Enzyme-linked immunosorbent assay; IGFBP2—Insulin-like growth factor-binding protein 2; IGFBP3—Insulin-like growth factor-binding protein 3; IHC—Immunohistochemistry; iTRAQ—Isobaric tags for relative and absolute quantification; LRG1—Leucine-rich alpha-2 glycoprotein; MALDI-ToF-MS—Matrix-assisted laser desorption/ionisation-time of flight mass spectrometry; MMP-9—Matrix metalloproteinase-9; MRM—Multiple reaction monitoring; RPLC—Reversed-phase liquid chromatography; RPPA—Reverse-phase protein array; SID MRM-MS—Stable isotope dilution multiple reaction monitoring mass spectrometry; THBS2—Thrombospondin-2; TIMP-1—Tissue inhibitor of metalloproteinase 1; TTR—Transthyretin.* The specificity and sensitivity of the biomarkers reported were based on the results of validation studies.

**Table 3 ijms-23-02093-t003:** Recently identified prognostic biomarkers for PDAC using proteomics techniques.

Name	Samples	Proteomic Techniques	Validation			References
			Method	Sensitivity *	Specificity *	
AGP1	Tissues	LC-MS/MS	PRM and IHC	Not reported	Not reported	[186]
Fibrinogen	Serum	MALDI-ToF MS	ELISA	67%	84%	[192,193]
H1.3	Tissues	LC-MS/MS	IHC	Not reported	Not reported	[185]
PNMAL1	Tissues	LC-MS/MS	IHC	Not reported	Not reported	[175]
Survivin	Tissues		IHC	Not reported	Not reported	[184]

AGP1—Alpha-1-acid glycoprotein 1; ELISA—Enzyme-linked immunosorbent assay; IHC—Immunohistochemistry; LC-MS/MS—Liquid chromatography tandem mass spectrometry; MALDI-ToF-MS—Matrix-assisted laser desorption/ionisation-time of flight mass spectrometry; PNMAL1—Paraneoplastic Ma antigen–like 1; PRM—Parallel reaction monitoring. * The specificity and sensitivity of the biomarkers reported were based on the results of validation studies.

**Table 4 ijms-23-02093-t004:** Recently identified biomarker for monitoring treatment response and tumour recurrence of PDAC patients using proteomics techniques.

Name	Samples	Proteomic Techniques	Validation			References
			Method	Sensitivity *	Specificity *	
Monitoring treatment response						
Panel of PZ, VWF, in combination with CA 19-9	Plasma	LC-MS/MS	ELISA	90%	61%	[208]
Monitoring tumour recurrence						
Galectin 4	Tissues	LC-MS/MS	Yes: PRM	Not reported	Not reported	[212]

CA 19-9—Cancer antigen 19-9; ELISA—Enzyme-linked immunosorbent assay; LC-MS/MS—Liquid chromatography tandem mass spectrometry; PRM—Parallel reaction monitoring; PZ—Vitamin-K dependent protein Z; vWF—von Willebrand factor. * The specificity and sensitivity of the biomarkers reported were based on the results of validation studies.

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
