# Peer review of "Application of Proteomics in Pancreatic Ductal Adenocarcinoma Biomarker Investigations: A Review"

_ijms, 2022, doi:10.3390/ijms23042093_

Round 1

Reviewer 1 Report

The review is aimed to describe the current views on proteomic approaches for biomarker discovery in the pancreatic ductal adenocarcinoma patients. The manuscript comprehensively describes the pathophysiology of PDAC and different proteomic-based approaches for biomarker discovery. The main strength of the review is in-depth summary of PDAC biomarkers for early detection or diagnosis, tumor recurrence, and monitoring treatment response. Overall, the review is interesting, well-written, and merits a publication.

Author Response

We thank Reviewer 1 for the positive feedback and comments

Reviewer 2 Report

The manuscript by Vellan and co-workers is an interesting overview about the literature data on the relationship between protein expression and early stage detection of pancreatic ductal adenocarcinoma.

The topic of the manuscript well fits with the scope of IJMS and is of interest for scientists working in the field. The overall evaluation of this reviewer is positive, although some revisions are suggested before further processing the paper.

In details:

  • Paragraphs should be numbered
  • Lines 56-58. Authors reported statics about Malaysia, Japan and Asian countries. But at line 65 they claim about myriad of researches about biomarkers. This reviewer would suggest to take into consideration the worldwide distribution of PanC.
  • Authors inserted a paragraph about PDAC treatment. Although this reviewer understands that this can be useful for the flux of whole story, a better integration of this part with the main topic of the review is suggested.
  • Authors inserted a paragraph titled “FDA-approved biomarkers for PDAC” (line 173). Here, they should insert more information about the FDA approval and clinical use, while in its current form only information about the usefulness of cancer antigen 1 and carcinoembryonic antigen are reported. The paragraph needs to be improved
  • Lines 196-229. Here, general information about proteomics techniques are reported. This part should be shortened
  • In paragraphs under the main title biomarker investigations, the organization "one-work one-paragraph should be avoided with the different parts better connected each others.
  • Please check the position of tables 2-4
  • In the last section, can authors summarize the different biomarkers in terms of efficiency and score the success rate of each group of biomarkers?

Author Response

Point 1:The manuscript by Vellan and co-workers is an interesting overview about the literature data on the relationship between protein expression and early-stage detection of pancreatic ductal adenocarcinoma.The topic of the manuscript well fits with the scope of IJMS and is of interest for scientists working in the field. The overall evaluation of this reviewer is positive, although some revisions are suggested before further processing the paper.
Response 1:We thank Reviewer 2 for the positive feedback and comments. We have now revised the manuscript according to the comments and/or suggestions.

Point 2:Paragraphs should be numbered.
Response 2:The sections and sub-sections are now numbered as recommended. Thank you.

Point 3:Lines 56-58. Authors reported statics about Malaysia, Japan and Asian countries. But at line 65 they claim about myriad of researches about biomarkers. This reviewer would suggest to take into consideration the worldwide distribution of PanC.
Response 3:The worldwide distribution of PanC in terms of incidences and mortality rate may be found at Lines 56 – 57 (Page 2).Thank you.

Point 4:Authors inserted a paragraph about PDAC treatment. Although this reviewer understands that this can be useful for the flux of whole story, a better integration of this part with the main topic of the review is suggested.
Response 4:Thank you for the suggestion. We have nowadded a statement to highlight the potential use of protein-based biomarkers for guiding and administering personalised treatment(s) and care in patients (Lines 171-
174 (Pages 6-7); 618 – 621 (Page 17)).

Point 5:Authors inserted a paragraph titled “FDA-approved biomarkers for PDAC” (line 173). Here, they should insert more information about the FDA approval and clinical use, while in its current form only information about the usefulness of cancer antigen 1 and carcinoembryonic antigen are reported. The paragraph needs to be improved.
Response 5:Thank you for highlighting this. We have now included relevant information on the subject in the newly revised manuscript as suggested (Lines 177-186; Lines 199-203; Page 7).

Point 6:Lines 196-229. Here, general information about proteomics techniques is reported. This part should be shortened.
Response 6:This section is now summarised as recommended(Lines 211 – 229, Pages 7-8). Thank you. 

Point 7:In paragraphs under the main title biomarker investigations, the organization "one-work one-paragraph should be avoided with the different parts better connected each other.
Response 7:We thank Reviewer 2 for the comment. While we agree with Reviewer 2 on this point to a certainextent, we would like to retain the current (or existing) flow of the manuscript as it has been designed and intended for such a presentation so that each of the selected recently identified biomarker(s) are described methodically in terms of the proteomics technology used for its discovery, the results obtained, its rates of efficiency as well as supporting evidences relating to its potential use (or otherwise). We believe that this manner of a presentation would equally benefit readers both novice and experts in the field of proteomics and biomarkers, as they could instantaneously obtain pertinent information regarding a particular biomarker as well as the various applications of proteomics technologies in cancer biomarker research.

Point 8:Please check the position of tables 2-4.
Response 8:The positioning of Tables 2 - 4 has now been checked and placed appropriately in the newly revised manuscript.Thank you.

Point 9:In the last section, can authors summarize the different biomarkers in terms of efficiency and score the success rate of each group of biomarkers?
Response 9:We thank Reviewer 2 for the excellent suggestion. We have now included additional information about the performance of each biomarkers (if available) in terms of its specificity and sensitivity for PDAC within Tables 2 - 4 for improved clarity and understanding. 

Round 2

Reviewer 2 Report

Authors addressed all the previous comments and provided suitable replies. The manuscript was revised taking into account the suggestions and can be now recommended for publication in IJMS in its current form